# The Consistency of Estimators in a Heteroscedastic Partially Linear Model with $\rho^-$-Mixing Errors

**Yu Zhang** and **Xinsheng Liu** *

State Key Laboratory of Mechanics and Control of Mechanical Structures and Department of Mathematics, Nanjing University of Aeronautics and Astronautics, Nanjing 210016, China; yuzhang@nuaa.edu.cn
* Correspondence: xsliu@nuaa.edu.cn

**Abstract:** This paper studies a heteroscedastic partially linear model based on $\rho^-$-mixing random errors, stochastically dominated and with zero mean. Under some suitable conditions, the strong consistency and $p$-th $(p > 0)$ mean consistency of least squares (LS) estimators and weighted least squares (WLS) estimators for the unknown parameter are investigated, and the strong consistency and $p$-th $(p > 0)$ mean consistency of the estimators for the non-parametric component are also studied. These results include the corresponding ones of independent, negatively associated (NA), and $\rho^*$-mixing random errors as special cases. At last, two simulations are presented to support the theoretical results.

**Keywords:** $\rho^-$-mixing random variables; heteroscedastic; partially linear model; LS estimators; WLS estimators; strong consistency; $p$-th mean consistency

**MSC:** 62G05; 62G20; 62F12

## 1. Introduction

Consider the following heteroscedastic partially linear model:

$$y^{(t)}(x_{in}, z_{in}) = z_{in}\beta + h(x_{in}) + \sigma_{in}\varepsilon^{(t)}(x_{in}), \ 1 \le t \le r, \ 1 \le i \le n, \tag{1}$$

where $\sigma_{in}^2 = f(u_{in})$, $z_{in} \in R$, $x_{in} \in R^p$, $u_{in} \in R^p$, and $(x_{in}, z_{in}, u_{in})$ are known and nonrandom design points, $\beta$ represents an unknown parameter, $f(\cdot)$ and $h(\cdot)$ represent unknown functions, which are defined on a compact set $M \subset R^p$, $y^{(t)}(x_{in}, z_{in})$ stands for the $t$-th variables that can be observable at points $(x_{in}, z_{in})$, and $\left\{\varepsilon^{(t)}(x_{in}), 1 \le t \le r, 1 \le i \le n\right\}$ stands for random errors.

In order to analyze the effect of temperature on electricity usage, Engle et al. [1] proposed the partially linear model

$$y_i = x'_i\beta + h(z_i) + \varepsilon_i, \ 1 \le i \le n. \tag{2}$$

Since then, many statisticians have studied partially linear regression models. The model (2) was further investigated by Heckman [2], Speckman [3], Gao [4], Härdle et al. [5], Hu et al. [6], Zeng and Liu [7], and so forth. Some applications of the model were given. Inspired by the model (2), a more general model was proposed by Gao et al. [8]:

$$y_i = x_i\beta + h(z_i) + \sigma_i\varepsilon_i, \ 1 \le i \le n. \tag{3}$$

Gao et al. [8] established the asymptotic normality of least squares (LS) and weighted least squares (WLS) estimators for $\beta$ based on the family of non-parametric estimators for $h(\cdot)$ and $f(\cdot)$ in the model (3). Baek and Liang [9] investigated the asymptotic property in the model (3) for negatively associated

errors. Zhou et al. [10] derived the moment consistency for $\beta$ and $h(\cdot)$ in model (3) under negatively associated samples. Hu [11] proposed a new partially linear model

$$y^{(t)}(x_{in}, z_{in}) = z_{in}\beta + h(x_{in}) + \varepsilon^{(t)}(x_{in}), \ 1 \le t \le r, \ 1 \le i \le n, \tag{4}$$

and derived the strong and moment consistencies with independent and $\varphi$-mixing errors. Li and Yang [12,13] studied the moment and strong consistencies for $\beta$ and $h(\cdot)$ in the model (4) based on negatively associated samples. Wang et al. [14] and Wu and Wang [15] discussed the moment and strong consistencies for LS and WLS estimators of $\beta$ and $h(\cdot)$ with $\rho^*$-mixing errors. In the present paper, we will investigate the model (1) based on the model (4). The model (1) can be used in hydrology, biology, and so on (see [16]).

Now, let us recall some concepts of dependent structures. Assume that $\mathbb{N}$ is a set of natural numbers and $S, T \subset \mathbb{N}$ are two non-empty disjoint sets. We define that $\text{dist}(S, T) = \min\limits_{i \in S, j \in T} |i - j|$.

**Definition 1** ([17]). *A finite collection of random variables $\{X_n, n \ge 1\}$ is called negatively associated (NA) if for each pair of disjoint subsets $B_1, B_2$ of $\mathbb{N}$,*

$$Cov\{g_1(X_k, k \in B_1), g_2(X_l, l \in B_2)\} \le 0,$$

*where $g_1$ and $g_2$ are any coordinate-wise non-decreasing functions such that the covariance exists.*

**Definition 2** ([18]). *A random sequence $\{X_n, n \ge 1\}$ is said to be $\rho^*$-mixing if*

$$\rho^*(u) = \sup\{\rho(T, U) : T, U \subset \mathbb{N}, \text{dist}(T, U) \ge u\} \to 0 \, (u \to \infty),$$

*where*

$$\rho(T, U) = \sup\left\{ \frac{\left|E(XY) - E(X)E(Y)\right|}{\sqrt{Var(X)Var(Y)}} : X \in L_2(\sigma(T)), Y \in L_2(\sigma(U)) \right\},$$

*$\sigma(T)$ and $\sigma(U)$ are $\sigma$-fields that are generated by $\{X_j, j \in T\}$ and $\{X_k, k \in U\}$ respectively, $L_2(\sigma(T))$ is the space of all square integral and $\sigma(T)$-measurable random variables, and $L_2(\sigma(U))$ is defined in the same way.*

**Definition 3** ([19,20]). *A sequence $\{X_n, n \ge 1\}$ is said to be $\rho^-$-mixing if*

$$\rho^-(u) = \sup\{\rho^-(T, U) : T, U \subset \mathbb{N}, \text{dist}(T, U) \ge u\} \to 0 \, (u \to \infty),$$

*where*

$$\rho^-(T, U) = 0 \vee \left\{ \frac{Cov\Big(g_1(X_i, i \in T), g_2\big(X_j, j \in U\big)\Big)}{\sqrt{Var(g_1(X_i, i \in T))Var\big(g_2\big(X_j, j \in U\big)\big)}} : g_1, g_2 \in H \right\},$$

*$0 \vee x = \max\{0, x\}$, and $H$ is the set of non-decreasing functions.*

We can easily see that $\rho^-(u) \le \rho^*(u)$ and a $\rho^-$-mixing sequence is NA (in particular, independent) if and only if $\rho^-(1) = 0$. Therefore, $\rho^-$-mixing sequences include $\rho^*$-mixing sequences and NA sequences as special cases. However, $\rho^*$-mixing sequences and NA sequences are not always $\rho^-$-mixing sequences. Zhang and Wang [19] constructed the following example, which is a $\rho^-$-mixing sequence, but not NA and also not $\rho^*$-mixing.

**Example 1** ([19]). *Assume that* $\{\xi_n, n \geq 1\}$, $\{\eta_n, n \geq 1\}$, *and* $\{\zeta_n, n \geq 1\}$ *are three independent sequences that are independent and identically distributed (i.i.d.) standard normal random variables. Denote*

$$X_n = \begin{cases} \xi_t, n = 2t - 1 \\ -\xi_t, n = 2t \end{cases}, \; Y_n = \begin{cases} \eta_t, n = 2^{2t-1} \\ -\eta_t, n = 2^{2t} \\ \zeta_n, \text{ else} \end{cases},$$

*and* $Z_n = X_n^2 + Y_n$. *Then,* $\{Z_n, n \geq 1\}$ *is* $\rho^-$-*mixing with* $\rho^-(2) = 0$. *However,* $\{Z_n, n \geq 1\}$ *is neither NA nor* $\rho^*$-*mixing.*

On one hand, NA sequences have been widely applied to reliability theorem and multivariate statistical analysis (see [21,22]). On the other hand, some Markov Chains and moving average processes are $\rho^*$-mixing sequences (see [23]). The concept of $\rho^*$-mixing sequences is important in a lot of areas, for instance, finance, economics, and other sciences (see [24]). Therefore, studying $\rho^-$-mixing sequences is of considerable significance.

Since Zhang and Wang [19] proposed the concept of $\rho^-$-mixing sequences, many results on $\rho^-$-mixing sequences have been established. One can refer to Zhang and Wang [19], Wang and Lu [25], and Yuan and An [26] for some moment inequalities and some limiting behavior; Zhang [20] and Zhang [27] for some central limit theorems; Chen et al. [28] for complete convergence for weighted sums of $\rho^-$-mixing sequences; Zhang [29] for the complete moment convergence for the partial sum of $\rho^-$-mixing moving average processes; Wu and Jiang [30] for almost sure convergence of $\rho^-$-mixing sequences; and Xu and Wu [31] for an almost sure central limit theorem for the self-normalized partial sums.

However, we have not found studies on the model (1) under $\rho^-$-mixing random errors in the literature. In the present paper, we will study the estimation problem for the model (1) based on the assumption that the errors are $\rho^-$-mixing sequences that are stochastically dominated and zero mean. The strong consistency and mean consistency of LS estimators and WLS estimators for $\beta$ and $h(\cdot)$ are established respectively based on some suitable conditions. The results obtained in the paper deal with independent errors as well as dependent errors as special cases.

Next, we will recall the definition of stochastic domination.

**Definition 4** ([32]). *A random sequence* $\{Y_i, i \geq 1\}$ *is stochastically dominated by a random variable* $Y$ *if*

$$P(|Y_i| > y) \leq cP(|Y| > y)$$

*for some* $c > 0$, *every* $y \geq 0$ *and each* $n \geq 1$.

The remainder of this paper is organized as follows. The LS estimators and WLS estimators of $\beta$ based on the family of non-parametric estimators for $h(\cdot)$ and some conditions are introduced in Section 2. We give the main results in Section 3. Several lemmas are given in Section 4. We provide the proofs of the main results in Section 5. Two simulations are carried out in Section 6. We conclude the paper in Section 7. Throughout the paper, let $C$ denote positive constants whose values may be different in various places. "i.i.d." stands for independent and identically distributed. $\|\cdot\|$ stands for the Euclidean norm.

## 2. Estimation and Conditions

Assume that $\left\{y^{(t)}(x_{in}, z_{in}), z_{in} \in R, x_{in} \in M, u_{in} \in M, 1 \leq t \leq r, 1 \leq i \leq n\right\}$ satisfies the model (1) and $W_{nj}(x) = W_{nj}(x; x_1, x_2, \cdots, x_n)$ is a weight function that is measurable on the compact set $M$. For simplicity and convenience, the model (1) can be written as

$$y_i^{(t)} = z_i\beta + h(x_i) + \sigma_i\varepsilon_i^{(t)}, \; 1 \leq t \leq r, \; 1 \leq i \leq n. \tag{5}$$

We denote $\widetilde{z}_j = z_j - \sum\limits_{i=1}^{n} W_{ni}(x_j)z_i$, $\widetilde{y}_{(k)}^j = y_j^{(k)} - \frac{1}{r}\sum\limits_{t=1}^{r}\sum\limits_{i=1}^{n} W_{ni}(x_j)y_i^{(t)}$, $\gamma_i = 1/\sigma_i^2$, $1 \le k \le r$, $1 \le j \le n$, $\widetilde{T}_n^2 = \sum\limits_{i=1}^{n} \widetilde{z}_i^2$, and $\widetilde{U}_n^2 = \sum\limits_{i=1}^{n} \gamma_i\widetilde{z}_i^2$.

For the model (5), one can get from $E\left(\varepsilon_i^{(t)}\right) = 0$ that $h(x_i) = E\left(y_i^{(t)} - z_i\beta\right)$ for $1 \le t \le r$, $1 \le i \le n$. Thus, for any given $\beta$, we can define the non-parametric estimator of $h(\cdot)$ in terms of

$$h_{r,n}(x, \beta) = \frac{1}{r}\sum_{t=1}^{r}\sum_{i=1}^{n} W_{ni}(x)\left(y_i^{(t)} - z_i\beta\right). \tag{6}$$

Hence, the LS estimators of $\beta$ can be defined by

$$\hat{\beta}_{r,n}^{(LS)} = \arg\min_{\beta}\sum_{t=1}^{r}\sum_{i=1}^{n}\left[y_i^{(t)} - z_i\beta - h_{r,n}(x_i, \beta)\right]^2. \tag{7}$$

By (7), we have

$$\hat{\beta}_{r,n}^{(LS)} = \frac{1}{r}\sum_{t=1}^{r}\sum_{i=1}^{n}\widetilde{z}_i\widetilde{y}_i^{(t)}/\widetilde{T}_n^2. \tag{8}$$

When the random errors are heteroscedastic, we modify $\hat{\beta}_{r,n}^{(LS)}$ to a WLS estimator. We can define the WLS estimators of $\beta$ in terms of

$$\hat{\beta}_{r,n}^{(WLS)} = \arg\min_{\beta}\sum_{t=1}^{r}\sum_{i=1}^{n}\left[\left(y_i^{(t)} - z_i\beta - h_{r,n}(x_i, \beta)\right)/\sigma_i\right]^2. \tag{9}$$

By (9), we derive that

$$\hat{\beta}_{r,n}^{(WLS)} = \frac{1}{r}\sum_{t=1}^{r}\sum_{i=1}^{n}\gamma_i\widetilde{z}_i\widetilde{y}_i^{(t)}/\widetilde{U}_n^2. \tag{10}$$

Taking into account $\hat{\beta}_{r,n}^{(LS)}$ and $\hat{\beta}_{r,n}^{(WLS)}$, we define the estimator of $h(\cdot)$ respectively:

$$\hat{h}_{r,n}(x) = \frac{1}{r}\sum_{t=1}^{r}\sum_{i=1}^{n} W_{ni}(x)\left(y_i^{(t)} - z_i\hat{\beta}_{r,n}^{(LS)}\right) \tag{11}$$

and

$$\widetilde{h}_{r,n}(x) = \frac{1}{r}\sum_{t=1}^{r}\sum_{i=1}^{n} W_{ni}(x)\left(y_i^{(t)} - z_i\hat{\beta}_{r,n}^{(WLS)}\right). \tag{12}$$

In order to obtain the relevant theorems, several important conditions are given below.

($C_1$) (i) $\lim\limits_{n\to\infty} \widetilde{T}_n^2/n = C$;

(ii) $0 < s_0 \le \inf\limits_{u\in M} f(u) \le \sup\limits_{u\in M} f(u) \le S_0 < \infty$;

(iii) $f(\cdot)$ and $h(\cdot)$ are continuous functions on compact set $M$.

($C_2$) (i) $\sup\limits_{x\in M}\sum_{i=1}^{n}\left|W_{ni}(x)\right| = O(1)$;

(ii) $\sup\limits_{i\ge 1, x\in M}\left|W_{ni}(x)\right| = O(n^{-\alpha})$ for some $\alpha > 0$.

($C_3$) (i) $\sup\limits_{x\in M}\left|\sum_{i=1}^{n} W_{ni}(x) - 1\right| = o(1)$;

(ii) $\sup\limits_{x\in M}\sum_{i=1}^{n}\left|W_{ni}(x)\right|I(\|x_i - x\| > \delta) = o(1)$ for any $\delta > 0$.

$(C_4)$ $\sup\limits_{x \in M}\left|\sum_{i=1}^{n} W_{ni}(x)z_i\right| = O(1)$.

**Remark 1.** *Conditions $(C_1)(i)$ $(ii)$ are some regular conditions that are often imposed in studies of LS and WLS estimators in heteroscedastic partially linear models. One can refer to [5,8,9] and so on. $(C_1)$ $(iii)$ is mild and holds for most commonly used functions, such as polynomial and trigonometric functions (see [9]). Conditions $(C_2)$–$(C_4)$ are often applied to investigate strong consistency (see [9,33]) and mean consistency (see [10,16]). $(C_2)(ii)$ is weaker than the corresponding conditions of [16] and [33]. Thus, the above conditions are very mild. Moreover, by $(C_1)(i)$ $(ii)$, one can get that*

$$\widetilde{T}_n^{-2}\sum_{i=1}^{n}\left|\widetilde{z}_i\right| \leq C \tag{13}$$

*and*

$$\widetilde{U}_n^{-2}\sum_{i=1}^{n}\left|\gamma_i\widetilde{z}_i\right| \leq C. \tag{14}$$

## 3. Main Results

In this paper, let $\left\{\varepsilon_i^{(t)}, 1 \leq t \leq r, 1 \leq i \leq n\right\}$ be a $\rho^{-}$-mixing sequence with zero mean, which is stochastically dominated by a random variable $\varepsilon$.

**Theorem 1.** *Assume that $(C_1)$-$(C_3)$ hold. If $E|\varepsilon|^p < \infty$ for some $p > 2$, then*

$$\hat{\beta}_{r,n}^{(LS)} \overset{a.s.}{\rightarrow} \beta \tag{15}$$

*as $\min(r,n) \rightarrow \infty$ and*

$$\hat{\beta}_{r,n}^{(WLS)} \overset{a.s.}{\rightarrow} \beta. \tag{16}$$

*as $\min(r,n) \rightarrow \infty$.*

**Theorem 2.** *Under the conditions of Theorem 1, in addition, if $(C_4)$ holds, then*

$$\sup\limits_{x \in M}\left|\hat{h}_{r,n}(x) - h(x)\right| \overset{a.s.}{\rightarrow} 0 \tag{17}$$

*as $\min(r,n) \rightarrow \infty$ and*

$$\sup\limits_{x \in M}\left|\widetilde{h}_{r,n}(x) - h(x)\right| \overset{a.s.}{\rightarrow} 0. \tag{18}$$

*as $\min(r,n) \rightarrow \infty$.*

**Theorem 3.** *Assume that $(C_1)$-$(C_3)$ holds. If $E|\varepsilon|^p < \infty$ for some $p \geq 2$, then*

$$\lim\limits_{\min(r,n)\rightarrow\infty} E\left|\hat{\beta}_{r,n}^{(LS)} - \beta\right|^p = 0 \tag{19}$$

*and*

$$\lim\limits_{\min(r,n)\rightarrow\infty} E\left|\hat{\beta}_{r,n}^{(WLS)} - \beta\right|^p = 0. \tag{20}$$

**Theorem 4.** *Under the conditions of Theorem 3, in addition, if $(C_4)$ holds, then*

$$\lim\limits_{\min(r,n)\rightarrow\infty}\sup\limits_{x \in M} E\left|\hat{h}_{r,n}(x) - h(x)\right|^p = 0 \tag{21}$$

*and*

$$\lim_{\min(r,n)\to\infty} \sup_{x\in M} E\left|\widetilde{h}_{r,n}(x) - h(x)\right|^p = 0. \tag{22}$$

**Remark 2.** *Since $\rho^-$-mixing sequences include NA (in particular, independent) and $\rho^*$-mixing sequences, Theorems 1–4 also apply for NA and $\rho^*$-mixing sequences.*

## 4. Some Lemmas

From the definition of $\rho^-$-mixing sequences, we can get the first lemma.

**Lemma 1.** *If $\{X_i, i \geq 1\}$ is a $\rho^-$-mixing sequence with mixing coefficients $\rho^-(u)$, then $\{f_i(X_i), i \geq 1\}$ is still a $\rho^-$-mixing sequence with mixing coefficients not greater than $\rho^-(u)$. Here, $f_1, f_2, \cdots$ are non-decreasing functions (non-increasing functions).*

**Lemma 2** (Rosenthal-type inequality, [25,29]). *If $\{X_i, i \geq 1\}$ is a $\rho^-$-mixing sequence of zero mean with $E|X_i|^p < \infty$ for some $p \geq 2$, then there exists a constant $C > 0$ depending only on $p$ and $\rho^-(s)$ such that*

$$E\left(\max_{1\leq i\leq n}|S_i|^p\right) \leq C\left\{\sum_{i=1}^n E|X_i|^p + \left[\left(\sum_{i=1}^n EX_i^2\right)^{p/2}\right]^p\right\},$$

*for any $n \geq 1$, here $S_i = \sum_{j=1}^i X_j$.*

**Lemma 3** ([32]). *If $\{X_i, i \geq 1\}$ is a random sequence that is stochastically dominated by a random variable $X$, for every $a > 0$ and $\beta > 0$, we have*

$$E|X_i|^\beta I(|X_i| \leq a) \leq c_1\left[E|X|^\beta I(|X| \leq a) + a^\beta P(|X| > a)\right],$$

$$E|X_i|^\beta I(|X_i| > a) \leq c_2 E|X|^\beta I(|X| > a).$$

*Therefore,*

$$E|X_i|^\beta \leq cE|X|^\beta,$$

*where $c$ is a positive constant.*

**Lemma 4.** *Let $\left\{\varepsilon_i^{(t)}, 1 \leq t \leq r, 1 \leq i \leq n\right\}$ be a $\rho^-$-mixing sequence of zero mean. Suppose that $\{c_{ni}(v), 1 \leq i \leq n, n \geq 1\}$ is an array of functions defined on a compact set $M$ such that $\sup\limits_{v\in M}\sum\limits_{i=1}^n \left|c_{ni}(v)\right| = O(1)$ and $\sup\limits_{i\geq 1, v\in M} \left|c_{ni}(v)\right| = O(n^{-\gamma})$ for some $\gamma > 0$. If $E|\varepsilon|^p < \infty$ for some $p > 2$, then*

$$\sup_{v\in M}\left|\frac{1}{r}\sum_{t=1}^r \sum_{i=1}^n c_{ni}(v)\varepsilon_i^{(t)}\right| \overset{a.s.}{\to} 0 \tag{23}$$

*as $\min(r,n) \to \infty$.*

**Proof.** Denote

$$\varepsilon_{1i}^{(t)} = -r^{\frac{1}{p}}I\left(\varepsilon_i^{(t)} < -r^{\frac{1}{p}}\right) + \varepsilon_i^{(t)}I\left(\left|\varepsilon_i^{(t)}\right| \leq r^{\frac{1}{p}}\right) + r^{\frac{1}{p}}I\left(\varepsilon_i^{(t)} > r^{\frac{1}{p}}\right),$$

$$\varepsilon_{2i}^{(t)} = \varepsilon_i^{(t)} - \varepsilon_{1i}^{(t)} = \left(\varepsilon_i^{(t)} + r^{\frac{1}{p}}\right)I\left(\varepsilon_i^{(t)} < -r^{\frac{1}{p}}\right) + \left(\varepsilon_i^{(t)} - r^{\frac{1}{p}}\right)I\left(\varepsilon_i^{(t)} > r^{\frac{1}{p}}\right),$$

$$\varepsilon'^{(t)}_i = \varepsilon^{(t)}_{1i} - E\varepsilon^{(t)}_{1i}$$

and

$$\varepsilon''^{(t)}_i = \varepsilon^{(t)}_{2i} - E\varepsilon^{(t)}_{2i}.$$

Without loss of generality, one can suppose that $c_{ni}(v) > 0$. Hence, we know by Lemma 1 that $\left\{c_{ni}(v)\varepsilon^{(t)}_i, 1 \le t \le r, 1 \le i \le n\right\}$, $\left\{c_{ni}(v)\varepsilon'^{(t)}_i, 1 \le t \le r, 1 \le i \le n\right\}$, and $\left\{c_{ni}(v)\varepsilon''^{(t)}_i, 1 \le t \le r, 1 \le i \le n\right\}$ are also $\rho^-$-mixing sequences with zero mean. Note that $\varepsilon^{(t)}_i = \varepsilon'^{(t)}_i + \varepsilon''^{(t)}_i$. Hence, for any $v \in M$, we have

$$
\begin{aligned}
A_{r,n} &=: P\left(\left|\frac{1}{r}\sum_{t=1}^{r}\sum_{i=1}^{n} c_{ni}(v)\varepsilon^{(t)}_i\right| > \varepsilon\right) \\
&\le P\left(\left|\sum_{t=1}^{r}\sum_{i=1}^{n} c_{ni}(v)\varepsilon'^{(t)}_i\right| > r\varepsilon/2\right) + P\left(\left|\sum_{t=1}^{r}\sum_{i=1}^{n} c_{ni}(v)\varepsilon''^{(t)}_i\right| > r\varepsilon/2\right) \\
&=: \left(A^{(1)}_{r,n} + A^{(2)}_{r,n}\right).
\end{aligned}
\tag{24}
$$

The proof of (24) is similar to that of the Lemma 3.3 in Zhou and Hu [34]. By Lemma 3, we have $E\left|\varepsilon^{(t)}_i\right|^p \le CE|\varepsilon|^p < \infty$. Hence, for every $s > p > 2$, from the Markov inequality, Lemma 2 and $E|\varepsilon|^p < \infty$, we get that

$$
\begin{aligned}
A^{(1)}_{r,n} &\le Cr^{-s}E\left|\sum_{t=1}^{r}\sum_{i=1}^{n} c_{ni}(v)\varepsilon'^{(t)}_i\right|^s \\
&\le Cr^{-s}\left\{\sum_{t=1}^{r}\sum_{i=1}^{n} E\left|c_{ni}(v)\varepsilon'^{(t)}_i\right|^s + \left[\left(\sum_{t=1}^{r}\sum_{i=1}^{n} E\left(c_{ni}(v)\varepsilon'^{(t)}_i\right)^2\right)^{s/2}\right]\right\} \\
&\le Cr^{-s}\sup_{z\in M}\left\{\sum_{t=1}^{r}\sum_{i=1}^{n} E\left|c_{ni}(v)\varepsilon'^{(t)}_i\right|^s + \left[\left(\sum_{t=1}^{r}\sum_{i=1}^{n} E\left(c_{ni}(v)\varepsilon'^{(t)}_i\right)^2\right)^{s/2}\right]\right\} \\
&\le Cr^{-s}\left\{\sup_{z\in M}\sum_{t=1}^{r}\sum_{i=1}^{n} E\left|c_{ni}(v)\varepsilon'^{(t)}_i\right|^s + \left[\sup_{z\in M}\left(\sum_{t=1}^{r}\sum_{i=1}^{n} E\left(c_{ni}(v)\varepsilon'^{(t)}_i\right)^2\right)^{s/2}\right]\right\} \\
&\le Cr^{-s}\left(r^{s/p}n^{-\gamma(s-1)} + r^{s/2}n^{-\gamma s/2}\right) \le Cr^{-s/2}
\end{aligned}
\tag{25}
$$

and

$$
\begin{aligned}
A^{(2)}_{r,n} &\le Cr^{-p}E\left|\sum_{t=1}^{r}\sum_{i=1}^{n} c_{ni}(v)\varepsilon''^{(t)}_i\right|^p \\
&\le Cr^{-p}\left\{\sup_{z\in M}\sum_{t=1}^{r}\sum_{i=1}^{n} E\left|c_{ni}(v)\varepsilon''^{(t)}_i\right|^p + \left[\sup_{z\in M}\left(\sum_{t=1}^{r}\sum_{i=1}^{n} E\left(c_{ni}(v)\varepsilon''^{(t)}_i\right)^2\right)^{p/2}\right]\right\} \\
&\le Cr^{-p}\left(rn^{-\gamma(p-1)} + r^{p/2}n^{-\gamma p/2}\right) \\
&\le C\left(r^{-p+1} + r^{-p/2}\right).
\end{aligned}
\tag{26}
$$

Hence, it follows from (24) through (26) and $s > p > 2$ that

$$\sum_{r=1}^{\infty} A_{r,n} \le C\sum_{r=1}^{\infty}\left(r^{-s/2} + r^{-p+1} + r^{-p/2}\right) < \infty.$$

By the Borel–Cantenlli lemma, we obtain for any $v \in M$ that

$$\left|\frac{1}{r}\sum_{t=1}^{r}\sum_{i=1}^{n} c_{ni}(v)\varepsilon^{(t)}_i\right| \xrightarrow{a.s.} 0$$

as $\min(r, n) \to \infty$. Therefore, (23) follows. $\square$

**Lemma 5.** Let $\left\{\varepsilon_i^{(t)}, 1 \le t \le r, 1 \le i \le n\right\}$ be a $\rho^-$-mixing random sequence of zero mean. Suppose that $\{c_{ni}(v), 1 \le i \le n, n \ge 1\}$ is an array of functions defined on a compact set $M$ such that $\sup\limits_{v \in M} \sum\limits_{i=1}^{n} \left|c_{ni}(v)\right| = O(1)$ and $\sup\limits_{i \ge 1, v \in M} \left|c_{ni}(v)\right| = O(n^{-\alpha})$. If $E|\varepsilon|^p < \infty$ for some $p \ge 2$, then

$$\lim_{\min(r,n) \to \infty} \sup_{v \in M} E\left|\frac{1}{r}\sum_{t=1}^{r}\sum_{i=1}^{n} c_{ni}(v)\varepsilon_i^{(t)}\right|^p = 0. \tag{27}$$

**Proof.** Using the notations in the proof of Lemma 4 and by $C_p$ inequality (let $\{X_i, i \ge 1\}$ be a random sequence, then $E\left|\sum\limits_{i=1}^{n} X_i\right|^p \le n^{p-1} \sum\limits_{i=1}^{n} E|X_i|^p$ for $p > 1$), one gets that

$$E\left|\frac{1}{r}\sum_{t=1}^{r}\sum_{i=1}^{n} c_{ni}(v)\varepsilon_i^{(t)}\right|^p \le 2^{p-1}\left(E\left|\frac{1}{r}\sum_{t=1}^{r}\sum_{i=1}^{n} c_{ni}(v)\varepsilon'_i^{(t)}\right|^p + E\left|\frac{1}{r}\sum_{t=1}^{r}\sum_{i=1}^{n} c_{ni}(v)\varepsilon''_i^{(t)}\right|^p\right)$$
$$=: 2^{p-1}\left(G_{r,n}^{(1)} + G_{r,n}^{(2)}\right). \tag{28}$$

For every $s > p \ge 2$, by Lemma 2, Lemma 3, and $E|\varepsilon|^p < \infty$, we derived that

$$\begin{aligned}
\sup_{v \in M} G_{r,n}^{(1)} &\le \left(\sup_{v \in M} E\left|\frac{1}{r}\sum_{t=1}^{r}\sum_{i=1}^{n} c_{ni}(v)\varepsilon'_i^{(t)}\right|^s\right)^{p/s} \\
&\le C\left[\left(\frac{1}{r}\right)^s\left(\sup_{v \in M} E\left|\sum_{t=1}^{r}\sum_{i=1}^{n} c_{ni}(v)\varepsilon'_i^{(t)}\right|^s\right)\right]^{p/s} \\
&\le C\left(\frac{1}{r}\right)^p\left(\sup_{v \in M}\sum_{t=1}^{r}\sum_{i=1}^{n} E\left|c_{ni}(v)\varepsilon'_i^{(t)}\right|^s + \left[\sup_{v \in M}\left(\sum_{t=1}^{r}\sum_{i=1}^{n} E\left(c_{ni}(v)\varepsilon'_i^{(t)}\right)^2\right)\right]^{s/2}\right)^{p/s} \\
&\le C\left(\frac{1}{r}\right)^p\left(r^{s/p}n^{-\gamma(s-1)} + r^{s/2}n^{-\gamma s/2}\right)^{p/s} \\
&\le Cr^{-p/2}
\end{aligned} \tag{29}$$

and

$$\begin{aligned}
\sup_{v \in M} G_{r,n}^{(2)} &\le C\left(\frac{1}{r}\right)^p\left(\sup_{v \in M}\sum_{t=1}^{r}\sum_{i=1}^{n} E\left|c_{ni}(v)\varepsilon''_i^{(t)}\right|^p \right. \\
&\quad \left. + \left[\sup_{v \in M}\left(\sum_{t=1}^{r}\sum_{i=1}^{n} E\left(c_{ni}(v)\varepsilon'_i^{(t)}\right)^2\right)\right]^{p/2}\right) \\
&\le C\left(\frac{1}{r}\right)^p\left(rn^{-\gamma(p-1)} + r^{p/2}n^{-\gamma p/2}\right) \\
&\le Cr^{-p/2}.
\end{aligned} \tag{30}$$

Therefore, (27) follows from (28) through (30). □

## 5. Proofs of the Main Results

By (5), (8), and (10), we derive that

$$\hat{\beta}_{r,n}^{(LS)} - \beta = \widetilde{T}_n^{-2}\left(\frac{1}{r}\sum_{t=1}^{r}\sum_{i=1}^{n} \widetilde{z}_i\widetilde{e}_i^{(t)} + \sum_{i=1}^{n} \widetilde{z}_i\breve{h}(x_i)\right) \tag{31}$$

and

$$\hat{\beta}_{r,n}^{(WLS)} - \beta = \widetilde{U}_n^{-2}\left(\frac{1}{r}\sum_{t=1}^{r}\sum_{i=1}^{n} \gamma_i\widetilde{z}_i\widetilde{e}_i^{(t)} + \sum_{i=1}^{n} \gamma_i\widetilde{z}_i\breve{h}(x_i)\right), \tag{32}$$

where $\widetilde{e}_j^{(k)} = e_j^{(k)} - \frac{1}{r} \sum\limits_{t=1}^{r} \sum\limits_{i=1}^{n} W_{ni}(x_j)e_i^{(t)}$, $\widecheck{h}(x) = h(x) - \sum\limits_{j=1}^{n} W_{nj}(x)h(x_j)$ and $e_i^{(t)} = \sigma_i \varepsilon_i^{(t)}$, $1 \le k \le r$, $1 \le j \le n$.

**Proof of Theorem 1.** We only need to prove (16) since the proof of (15) is analogous. By (32), we can get that

$$
\begin{aligned}
\hat{\beta}_{r,n}^{(WLS)} - \beta &= \widetilde{U}_n^{-2}\left[\frac{1}{r}\sum_{t=1}^{r}\sum_{i=1}^{n}\gamma_i\widetilde{z}_i\sigma_i\varepsilon_i^{(t)} - \sum_{j=1}^{n}\gamma_j\widetilde{z}_j\left(\frac{1}{r}\sum_{t=1}^{r}\sum_{i=1}^{n}W_{ni}(x_j)\sigma_i\varepsilon_i^{(t)}\right) + \sum_{i=1}^{n}\gamma_i\widetilde{z}_i\widecheck{h}(x_i)\right] \\
&=: I_{r,n}^{(1)} - I_{r,n}^{(2)} + I_{r,n}^{(3)} \\
&\le \left|I_{r,n}^{(1)}\right| + \left|I_{r,n}^{(2)}\right| + \left|I_{r,n}^{(3)}\right|.
\end{aligned}
\tag{33}
$$

Observe that $I_{r,n}^{(1)} = \frac{1}{r}\sum\limits_{t=1}^{r}\sum\limits_{i=1}^{n}\left(\widetilde{U}_n^{-2}\gamma_i\sigma_i\widetilde{z}_i\right)\varepsilon_i^{(t)} \triangleq \frac{1}{r}\sum\limits_{t=1}^{r}\sum\limits_{i=1}^{n}a_{ni}\varepsilon_i^{(t)}$. Hence, it follows from $(C_1)$(i) and (ii) and (14) that

$$
\max_{1\le i\le n}|a_{ni}| \le C\max_{1\le i\le n}\left|\gamma_i\widetilde{z}_i\right|\widetilde{U}_n^{-1}\cdot\widetilde{U}_n^{-1} = O\left(n^{-1/2}\right)
\tag{34}
$$

and

$$
\sum_{i=1}^{n}|a_{ni}| \le C\sum_{i=1}^{n}\left|\gamma_i\widetilde{z}_i\right|\widetilde{U}_n^{-2} = O(1).
\tag{35}
$$

Thus, by Lemma 4, we have

$$
\left|I_{r,n}^{(1)}\right| \overset{a.s.}{\to} 0
\tag{36}
$$

as $\min(r,n) \to \infty$. Note that

$$
I_{r,n}^{(2)} = \frac{1}{r}\sum_{t=1}^{r}\sum_{i=1}^{n}\left(\sum_{j=1}^{n}\widetilde{U}_n^{-2}\gamma_j\sigma_i\widetilde{z}_jW_{ni}(x_j)\right)\varepsilon_i^{(t)} =: \frac{1}{r}\sum_{t=1}^{r}\sum_{i=1}^{n}a'_{ni}\varepsilon_i^{(t)}.
$$

Hence, it follows from $(C_1)$(ii), $(C_2)$, and (14) that

$$
\max_{1\le i\le n}\left|a'_{ni}\right| \le C\sup_{i\ge 1, x\in M}\left|W_{ni}(x)\right|\cdot\sum_{i=1}^{n}\left|\gamma_j\widetilde{z}_j\right|\widetilde{U}_n^{-2} = O(n^{-\alpha})
\tag{37}
$$

and

$$
\sum_{i=1}^{n}\left|a'_{ni}\right| \le C\sup_{x\in M}\sum_{i=1}^{n}\left|W_{ni}(x)\right|\cdot\sum_{i=1}^{n}\left|\gamma_j\widetilde{z}_j\right|\widetilde{U}^{-2} = O(1),
\tag{38}
$$

where $\alpha$ is the same as that in $(C_2)$(ii).

Thus, by Lemma 4, one can get that

$$
\left|I_{r,n}^{(2)}\right| \overset{a.s.}{\to} 0
\tag{39}
$$

as $\min(r,n) \to \infty$. By (14), we derive that

$$
\left|I_{r,n}^{(3)}\right| \le \sup_{x\in M}\left|\widecheck{h}(x)\right|\sum_{i=1}^{n}\left|\gamma_i z_i\right|/\widetilde{U}_n^2 \le C\sup_{x\in M}\left|\widecheck{h}(x)\right|.
\tag{40}
$$

By $(C_1)$(iii), $(C_2)$(i), and $(C_3)$, we obtain that

$$
\begin{aligned}
\sup_{x\in M}\left|\breve{h}(x)\right| \;\leq\; & \sup_{x\in M}\left|\sum_{j=1}^{n} W_{nj}(x)-1\right|\left|h(x)\right| + \sup_{x\in M}\sum_{j=1}^{n}\left|W_{nj}(x)\right|\left|h(x)-h(x_j)\right| \\
\leq\; & \sup_{x\in M}\left|\sum_{j=1}^{n} W_{nj}(x)-1\right|\left|h(x)\right| \\
& +\sup_{x\in M}\sum_{j=1}^{n}\left|W_{nj}(x)\right|\left|h(x)-h(x_j)\right|I\left(\|x-x_j\|>\delta\right) \\
& +\sup_{x\in M}\sum_{j=1}^{n}\left|W_{nj}(x)\right|\left|h(x)-h(x_j)\right|I\left(\|x-x_j\|\le\delta\right) \\
=\; & o(1).
\end{aligned}
\tag{41}
$$

Thus, by (40) and (41), we have

$$
\left|I_{r,n}^{(3)}\right|\to 0
\tag{42}
$$

as $\min(r,n)\to\infty$. Therefore, (16) follows from (33), (36), (39), and (42). □

**Proof of Theorem 2.** We only need to prove (18) since the proof of (17) is analogous. In light of (12), we have

$$
\begin{aligned}
\widetilde{h}_{r,n}(x)-h(x) \;=\; & \frac{1}{r}\sum_{t=1}^{r}\sum_{i=1}^{n} W_{ni}(x_i)\left(z_i\beta+h(x_i)+\sigma_i\varepsilon_i^{(t)}-z_i\hat{\beta}_{r,n}^{(WLS)}\right)-h(x) \\
=\; & \frac{1}{r}\sum_{t=1}^{r}\sum_{i=1}^{n} W_{ni}(x)z_i\left(\beta-\hat{\beta}_{r,n}^{(WLS)}\right)-\breve{h}(x)+\frac{1}{r}\sum_{t=1}^{r}\sum_{i=1}^{n} W_{ni}(x)\sigma_i\varepsilon_i^{(t)}.
\end{aligned}
\tag{43}
$$

Hence,

$$
\begin{aligned}
\sup_{x\in M}\left|\widetilde{h}_{r,n}(x)-h(x)\right| \;\leq\; & \sup_{x\in M}\left|\sum_{i=1}^{n} W_{ni}(x)z_i\right|\left|\beta-\hat{\beta}_{r,n}^{(WLS)}\right|+\sup_{x\in M}\left|\breve{h}(x)\right| \\
& +\sup_{x\in M}\left|\frac{1}{r}\sum_{t=1}^{r}\sum_{i=1}^{n} W_{ni}(x)\sigma_i\varepsilon_i^{(t)}\right| \\
=\; & : J_{r,n}^{(1)}+J_{r,n}^{(2)}+J_{r,n}^{(3)}.
\end{aligned}
\tag{44}
$$

By (16) and $(C_4)$, we have

$$
J_{r,n}^{(1)}\overset{a.s.}{\to}0
\tag{45}
$$

as $\min(r,n)\to\infty$. From (41), it follows that

$$
J_{r,n}^{(2)}\to 0
\tag{46}
$$

as $\min(r,n)\to\infty$. By $(C_1)$(ii) and $(C_2)$, we can get that

$$
\sup_{i\ge 1,x\in M}\left|W_{ni}(x)\sigma_i\right|\le C\sup_{i\ge 1,x\in M}\left|W_{ni}(x)\right|=O(n^{-\alpha}),
$$

and

$$
\sup_{x\in M}\sum_{i=1}^{n}\left|W_{ni}(x)\sigma_i\right|\le C\sup_{x\in M}\sum_{i=1}^{n}\left|W_{ni}(x)\right|=O(1).
$$

Hence, from Lemma 4, it follows that

$$
J_{r,n}^{(3)}\overset{a.s.}{\to}0.
\tag{47}
$$

as $\min(r,n)\to\infty$. Therefore, (18) follows from (44) through (47). □

**Proof of Theorem 3.** We only need to prove (20) since the proof of (19) is analogous. By (33), we have

$$
\begin{aligned}
\hat{\beta}_{r,n}^{(WLS)} - \beta \ &= \ \widetilde{U}_n^{-2}\left[\frac{1}{r}\sum_{t=1}^{r}\sum_{i=1}^{n}\gamma_i\widetilde{z}_i\sigma_i\varepsilon_i^{(t)} - \sum_{j=1}^{n}\gamma_j\widetilde{z}_j\left(\frac{1}{r}\sum_{t=1}^{r}\sum_{i=1}^{n}W_{ni}(x_j)\sigma_i\varepsilon_i^{(t)}\right) + \sum_{i=1}^{n}\gamma_i\widetilde{z}_i\breve{h}(x_i)\right] \\
&=: \ I_{r,n}^{(1)} - I_{r,n}^{(2)} + I_{r,n}^{(3)}.
\end{aligned}
$$

Hence, it follows by $C_p$ inequality that

$$
E\left|\hat{\beta}_{r,n}^{(WLS)} - \beta\right|^p \leq 3^{p-1}\left(E\left|I_{r,n}^{(1)}\right|^p + E\left|I_{r,n}^{(2)}\right|^p + E\left|I_{r,n}^{(3)}\right|^p\right). \tag{48}
$$

The rest of the proof is similar to the proof of (16), so we omitted the details here.　□

**Proof of Theorem 4.** We only need to prove (22) since the proof of (21) is analogous. By (43), we derive that

$$
\begin{aligned}
\widetilde{h}_{r,n}(x) - h(x) \ &= \ \frac{1}{r}\sum_{t=1}^{r}\sum_{i=1}^{n}W_{ni}(x)\left(z_i\beta + h(x) + \sigma_i\varepsilon_i^{(t)} - z_i\hat{\beta}_{r,n}^{(WLS)}\right) - h(x) \\
&= \ \frac{1}{r}\sum_{t=1}^{r}\sum_{i=1}^{n}W_{ni}(x)z_i\left(\beta - \hat{\beta}_{r,n}^{(WLS)}\right) - \breve{h}(x) + \frac{1}{r}\sum_{t=1}^{r}\sum_{i=1}^{n}W_{ni}(x)\sigma_i\varepsilon_i^{(t)} \\
&=: \ J_{r,n}^{(1)} - J_{r,n}^{(2)} + J_{r,n}^{(3)}.
\end{aligned}
$$

Hence, by $C_p$ inequality, we derive that

$$
E\left|\widetilde{h}_{r,n}(x) - h(x)\right|^p \leq 3^{p-1}\left(E\left|J_{r,n}^{(1)}\right|^p + E\left|J_{r,n}^{(2)}\right|^p + E\left|J_{r,n}^{(3)}\right|^p\right). \tag{49}
$$

Since $\left|J_{r,n}^{(1)}\right|^p \leq \left(\sup\limits_{x\in M}\left|\sum_{i=1}^{n}W_{ni}(x)z_i\right|\right)^p\left|\beta - \hat{\beta}_{r,n}^{(WLS)}\right|^p$, together with (20) and $(C_4)$, we can get that

$$
\lim_{\min(r,n)\to\infty}\sup_{x\in M}E\left|J_{r,n}^{(1)}\right|^p = 0. \tag{50}
$$

From (41), it follows that

$$
\lim_{\min(r,n)\to\infty}\sup_{x\in M}E\left|J_{r,n}^{(2)}\right|^p = 0. \tag{51}
$$

By $(C_1)$(ii) and $(C_2)$, we can get that

$$
\sup_{i\geq 1, x\in M}\left|W_{ni}(x)\sigma_i\right| \leq C\sup_{i\geq 1, x\in M}\left|W_{ni}(x)\right| = O(n^{-\alpha}),
$$

and

$$
\sup_{x\in M}\sum_{i=1}^{n}\left|W_{ni}(x)\sigma_i\right| \leq C\sup_{x\in M}\sum_{i=1}^{n}\left|W_{ni}(x)\right| = O(1).
$$

Hence, by Lemma 5, one can get that

$$
\lim_{\min(r,n)\to\infty}\sup_{x\in M}E\left|J_{r,n}^{(3)}\right|^p = 0. \tag{52}
$$

Therefore, (22) follows from (49) through (52).　□

## 6. Numerical Simulations

In this section, we will verify the validity of the theoretical results by two simulations.

*6.1. Simulation 1*

We will simulate a partially linear model

$$y_i^{(t)} = z_i\beta + h(x_i) + \sigma_i\varepsilon_i^{(t)}, \; r = n/2, 1 \le i \le n, \tag{53}$$

where $\beta = 2.5$, $h(x) = \sin(2\pi x)$, $z_i = (-1)^i \cdot \frac{i}{n}$, $\sigma_i = 1$, $1 \le i \le n$, and random errors $\left\{\varepsilon_i^{(t)}\right\}$ have the common distribution as that of $\{Z_n\}$ in Example 1 of Section 1. Then, $\left\{\varepsilon_i^{(t)}\right\}$ is a $\rho^-$-mixing sequence, and it is neither NA nor $\rho^*$-mixing.

In particular, we take the weight function $W_{ni}(\cdot)$ as the following nearest neighbor weight function (see [11,35]). Without loss of generality, denote $M = [0,1]$ and $x_i = \frac{i}{n}$ $(1 \le i \le n)$. For each $x \in M$, we rewrite

$$|x_1 - x|, |x_2 - x|, \cdots, |x_n - x|$$

as follows:

$$\left|x_{R_1(x)} - x\right| \le \left|x_{R_2(x)} - x\right| \le, \cdots, \le \left|x_{R_n(x)} - x\right|.$$

Take $k_n = \left[n^{0.8}\right]$ and define the nearest neighbor weight function as

$$W_{ni}(x) = \begin{cases} \frac{1}{k_n}, & \text{if} |x_i - x| \le \left|x_{R_{k_n}} - x\right|, \\ 0, & \text{else.} \end{cases}$$

where the sample sizes are taken as $n = 100, 600, 1200, 1900, 2700$, and 3600 and the points $x$ are taken as $x = 0.2, 0.4, 0.6$, and 0.8, respectively. We compute $\hat{\beta}_{r,n}^{(LS)} - \beta$ and $\hat{h}_{r,n}(x) - h(x)$ for 1000 times, respectively. The boxplots of $\hat{\beta}_{r,n}^{(LS)} - \beta$ are provided in Figures 1–4, the violin plots of $\hat{h}_{r,n}(x) - h(x)$ are provided in Figures 5–8, the curves of $h(x)$ and $\hat{h}_{r,n}(x)$ are provided in Figure 9, and the mean squared errors (MSE) of $\hat{\beta}_{r,n}^{(LS)}$ and $\hat{h}_{r,n}(x)$ are presented in Tables 1 and 2, respectively.

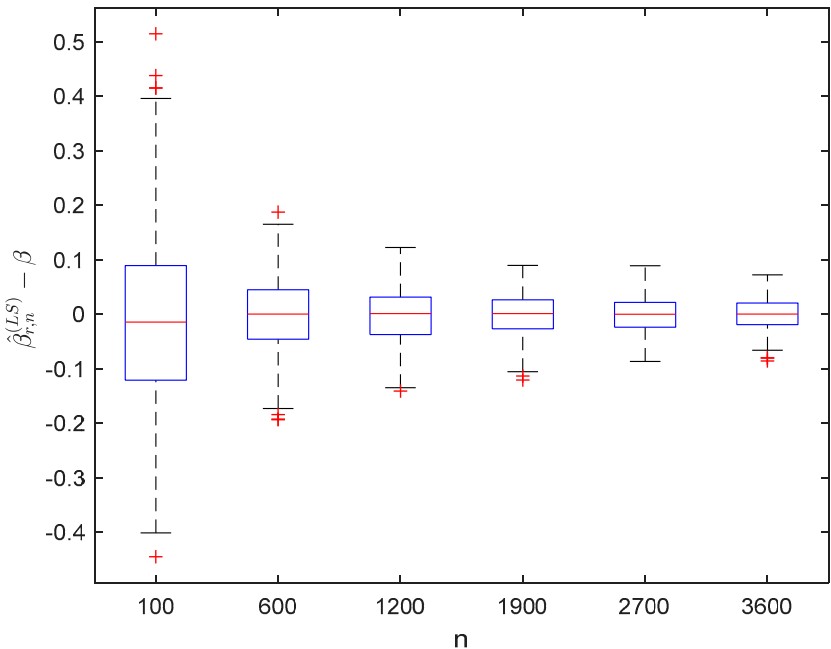

**Figure 1.** Boxplots of $\hat{\beta}_{r,n}^{(LS)} - \beta$ with $\beta = 2.5$ and $x = 0.2$.

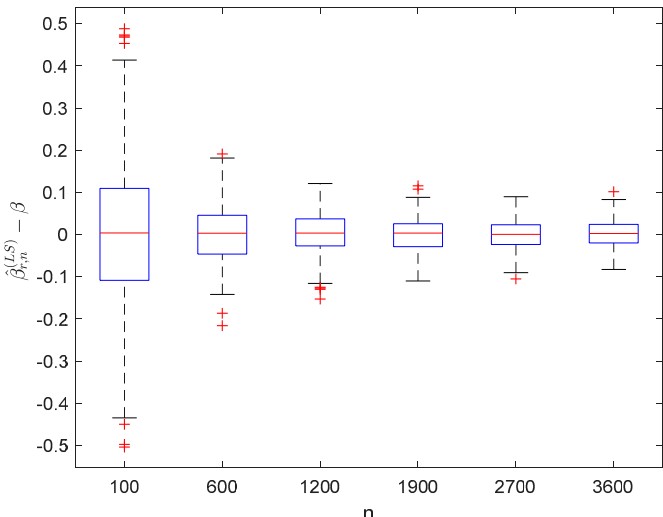

**Figure 2.** Boxplots of $\hat{\beta}_{r,n}^{(LS)} - \beta$ with $\beta = 2.5$ and $x = 0.4$.

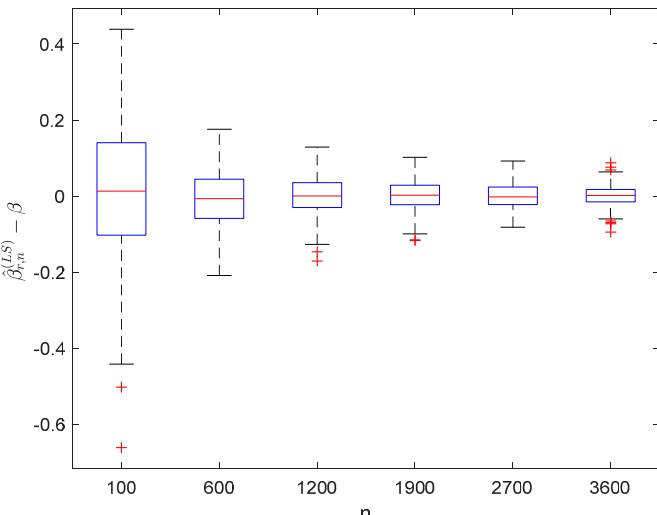

**Figure 3.** Boxplots of $\hat{\beta}_{r,n}^{(LS)} - \beta$ with $\beta = 2.5$ and $x = 0.6$.

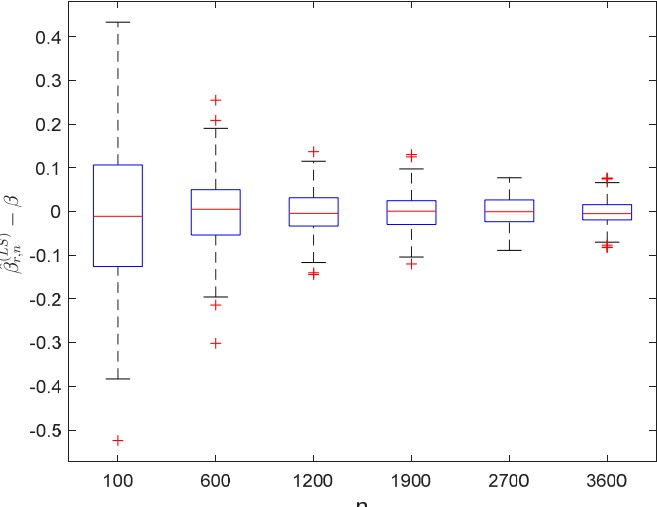

**Figure 4.** Boxplots of $\hat{\beta}_{r,n}^{(LS)} - \beta$ with $\beta = 2.5$ and $x = 0.8$.

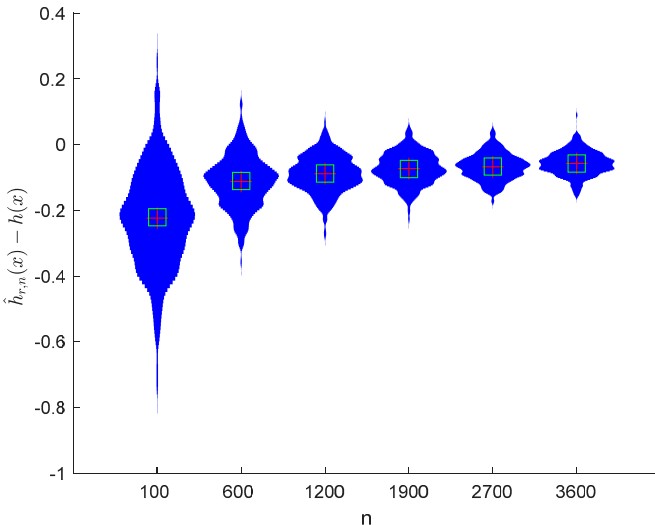

**Figure 5.** Violin plots of $\hat{h}_{r,n}(x) - h(x)$ with $\beta = 2.5$ and $x = 0.2$.

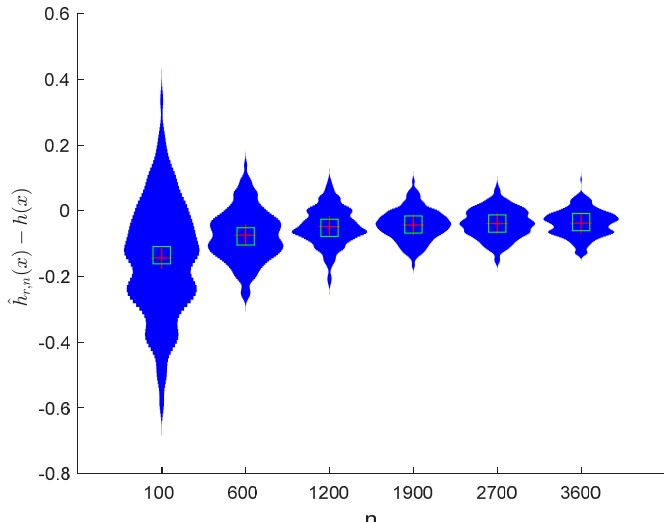

**Figure 6.** Violin plots of $\hat{h}_{r,n}(x) - h(x)$ with $\beta = 2.5$ and $x = 0.4$.

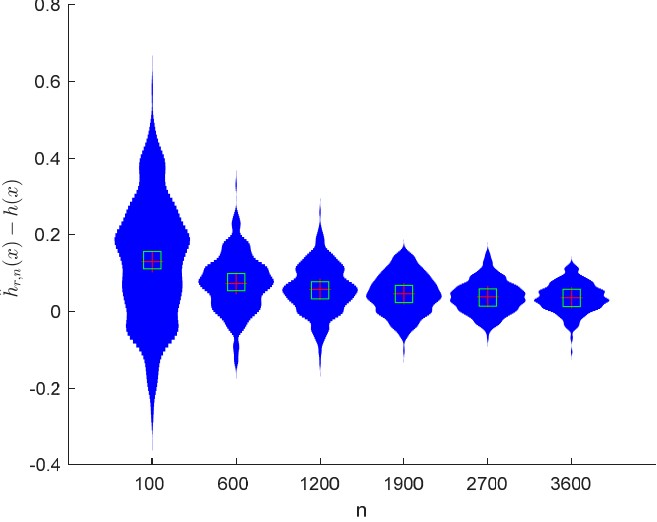

**Figure 7.** Violin plots of $\hat{h}_{r,n}(x) - h(x)$ with $\beta = 2.5$ and $x = 0.6$.

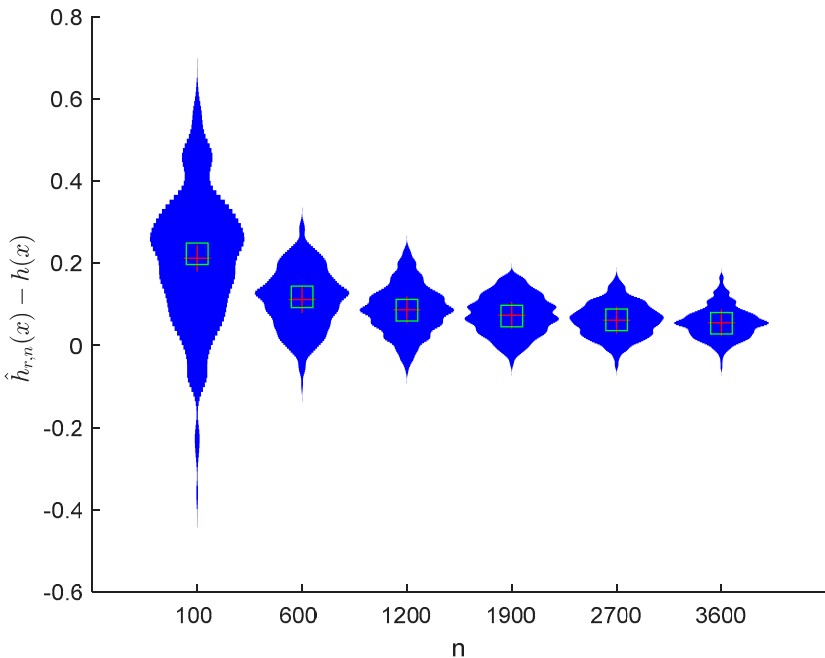

**Figure 8.** Violin plots of $\hat{h}_{r,n}(x) - h(x)$ with $\beta = 2.5$ and $x = 0.8$.

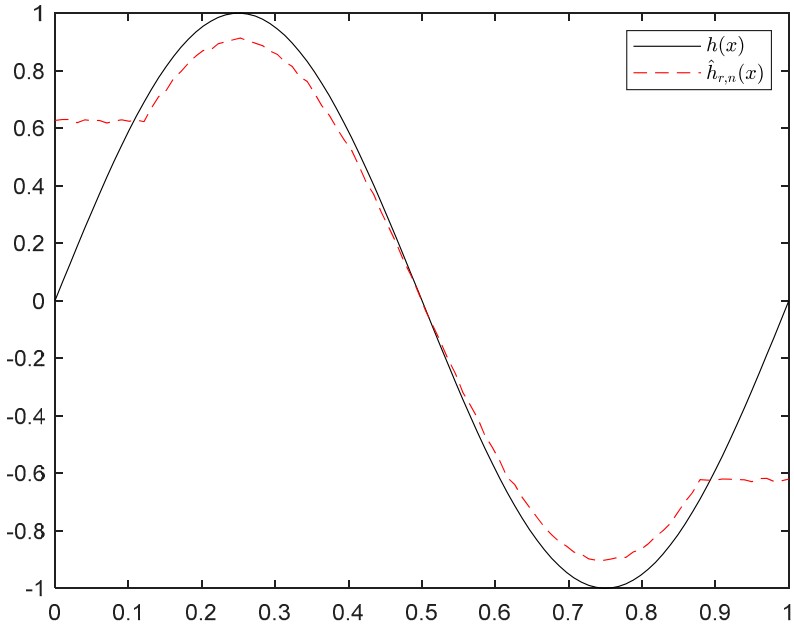

**Figure 9.** Curves of $h(x) = \sin(2\pi x)$ and $\hat{h}_{r,n}(x)$ with $\beta = 2.5$ and $n = 1200$.

**Table 1.** The MSEs of $\hat{\beta}_{r,n}^{(LS)}$ with $\beta = 2.5$ and $h(x) = \sin(2\pi x)$.

| $x$ | $n = 100$ | $n = 600$ | $n = 1200$ | $n = 1900$ | $n = 2700$ | $n = 3600$ |
|-----|-----------|-----------|------------|------------|------------|------------|
| 0.2 | 0.029564  | 0.0040429 | 0.0027393  | 0.0017879  | 0.0011778  | 0.00096205 |
| 0.4 | 0.027563  | 0.0048107 | 0.002557   | 0.0013618  | 0.0012225  | 0.00083697 |
| 0.6 | 0.032418  | 0.0045437 | 0.0030247  | 0.0017816  | 0.001016   | 0.0007695  |
| 0.8 | 0.026715  | 0.0049648 | 0.0026033  | 0.0014911  | 0.00099622 | 0.00083525 |

**Table 2.** The MSEs of $\hat{h}_{r,n}(x)$ with $\beta = 2.5$ and $h(x) = \sin(2\pi x)$.

| $x$ | $n = 100$ | $n = 600$ | $n = 1200$ | $n = 1900$ | $n = 2700$ | $n = 3600$ |
|-----|-----------|-----------|------------|------------|------------|------------|
| 0.2 | 0.072899 | 0.0132846 | 0.0064122 | 0.00450755 | 0.00408198 | 0.00309342 |
| 0.4 | 0.064407 | 0.012605 | 0.0061076 | 0.0040173 | 0.0037297 | 0.0027017 |
| 0.6 | 0.0651945 | 0.01143844 | 0.0061647 | 0.0048246 | 0.0031893 | 0.002914 |
| 0.8 | 0.067254 | 0.0134688 | 0.0701644 | 0.00515616 | 0.00397691 | 0.0030646 |

*6.2. Simulation 2*

We will simulate a partially linear model

$$y_i^{(t)} = z_i\beta + h(x_i) + \sigma_i\varepsilon_i^{(t)}, \; r = n/2, 1 \le i \le n, \tag{54}$$

where $\beta = 3.5$, $h(x) = \cos(\pi x)$, $z_i = (-1)^i \cdot \frac{i}{n}$, $\sigma_i = 1$, $1 \le i \le n$, and random errors $\left\{\varepsilon_i^{(t)}\right\}$ have the same distribution as $\{Z_n\}$ in Example 1 of Section 1. Then, $\left\{\varepsilon_i^{(t)}\right\}$ is a $\rho^-$-mixing sequence, and it is neither NA nor $\rho^*$-mixing.

Using the same estimating methods as model (53), we compute $\hat{\beta}_{r,n}^{(LS)} - \beta$ and $\hat{h}_{r,n}(x) - h(x)$ for 1000 times in model (54) under different values of $n$, respectively. The boxplots of $\hat{\beta}_{r,n}^{(LS)} - \beta$ are provided in Figures 10–13, the violin plots of $\hat{h}_{r,n}(x) - h(x)$ are provided in Figures 14–17, the curves of $h(x)$ and $\hat{h}_{r,n}(x)$ are provided in Figure 18, and the MSEs of $\hat{\beta}_{r,n}^{(LS)}$ and $\hat{h}_{r,n}(x)$ are presented in Tables 3 and 4, respectively.

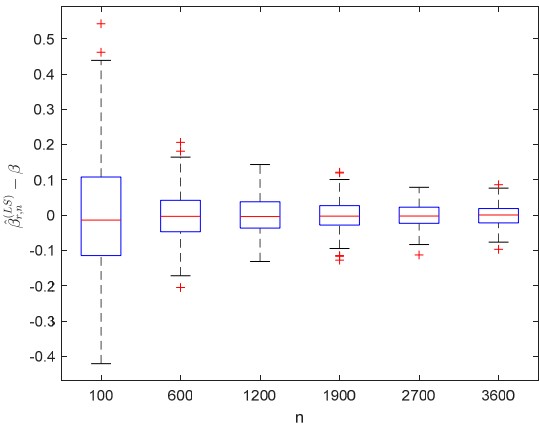

**Figure 10.** Boxplots of $\hat{\beta}_{r,n}^{(LS)} - \beta$ with $\beta = 3.5$ and $x = 0.2$.

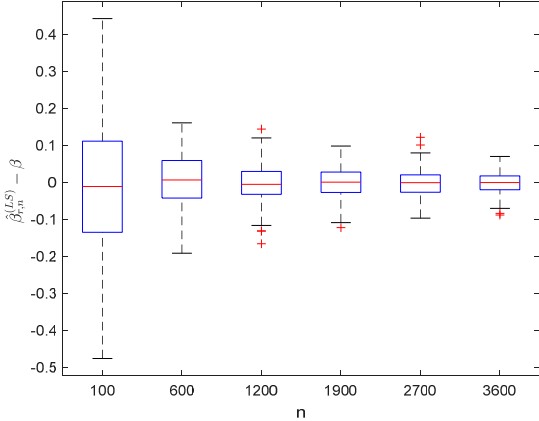

**Figure 11.** Boxplots of $\hat{\beta}_{r,n}^{(LS)} - \beta$ with $\beta = 3.5$ and $x = 0.4$.

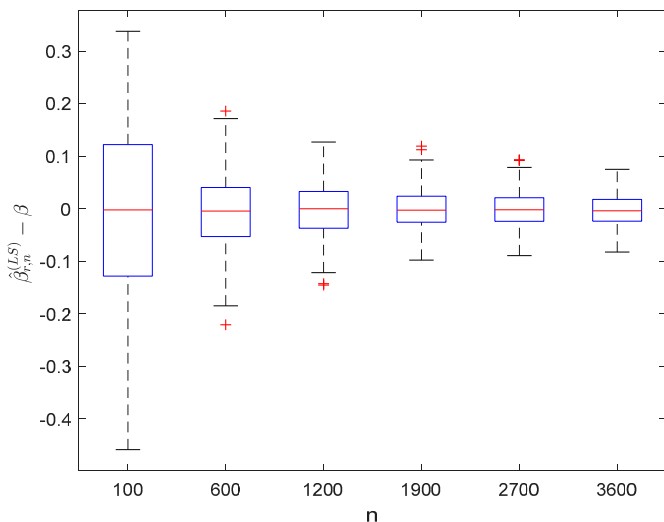

**Figure 12.** Boxplots of $\hat{\beta}_{r,n}^{(LS)} - \beta$ with $\beta = 3.5$ and $x = 0.6$.

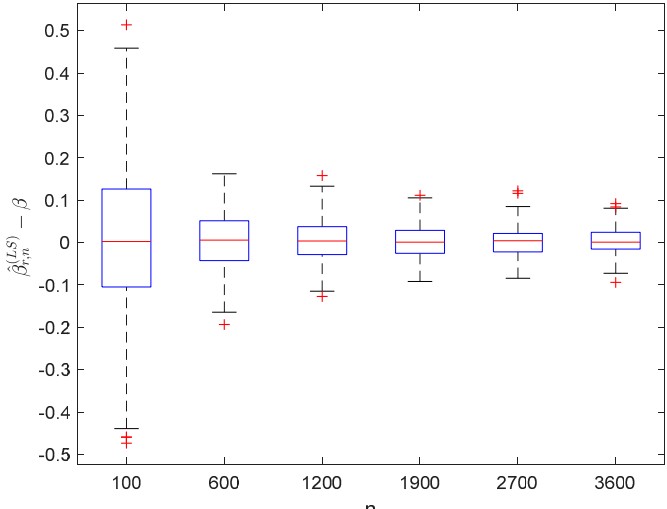

**Figure 13.** Boxplots of $\hat{\beta}_{r,n}^{(LS)} - \beta$ with $\beta = 3.5$ and $x = 0.8$.

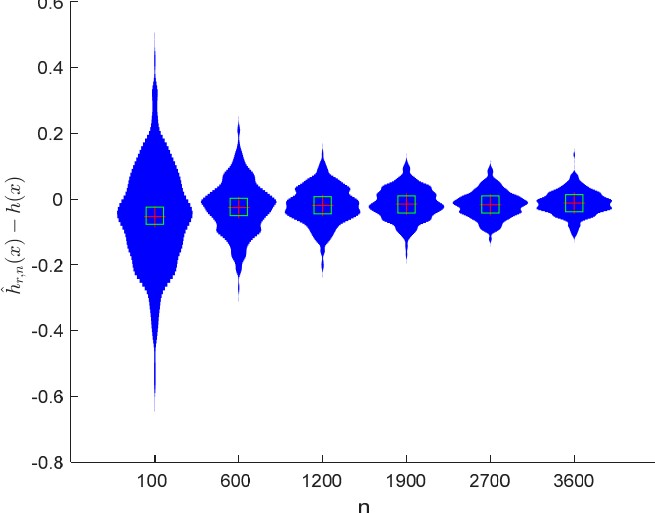

**Figure 14.** Violin plots of $\hat{h}_{r,n}(x) - h(x)$ with $\beta = 3.5$ and $x = 0.2$.

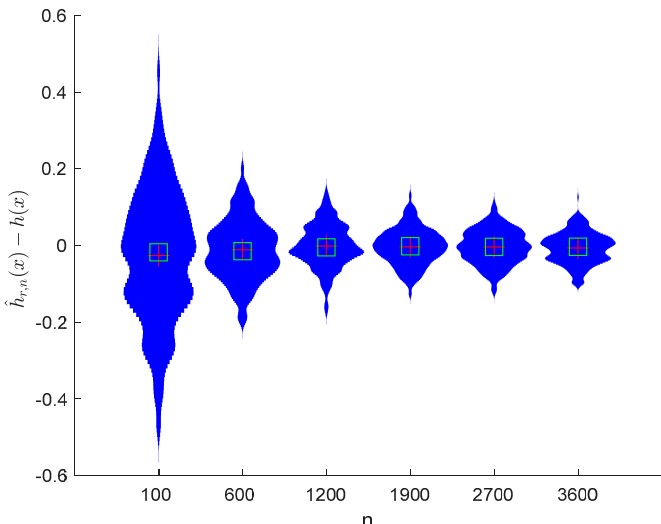

**Figure 15.** Violin plots of $\hat{h}_{r,n}(x) - h(x)$ with $\beta = 3.5$ and $x = 0.4$.

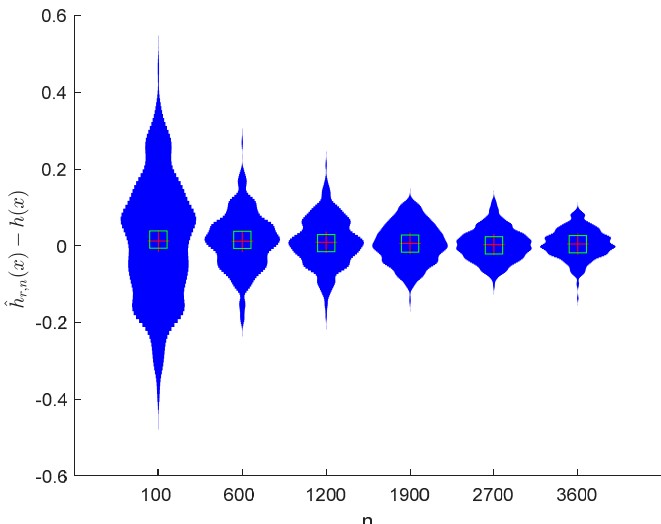

**Figure 16.** Violin plots of $\hat{h}_{r,n}(x) - h(x)$ with $\beta = 3.5$ and $x = 0.6$.

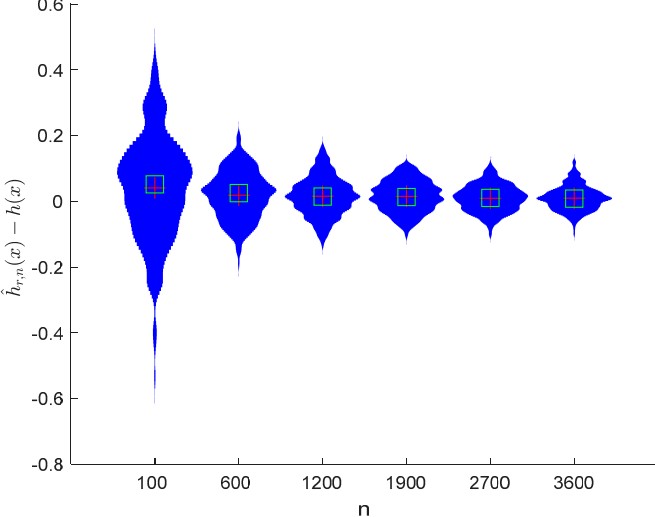

**Figure 17.** Violin plots $\hat{h}_{r,n}(x) - h(x)$ with $\beta = 3.5$ and $x = 0.8$.

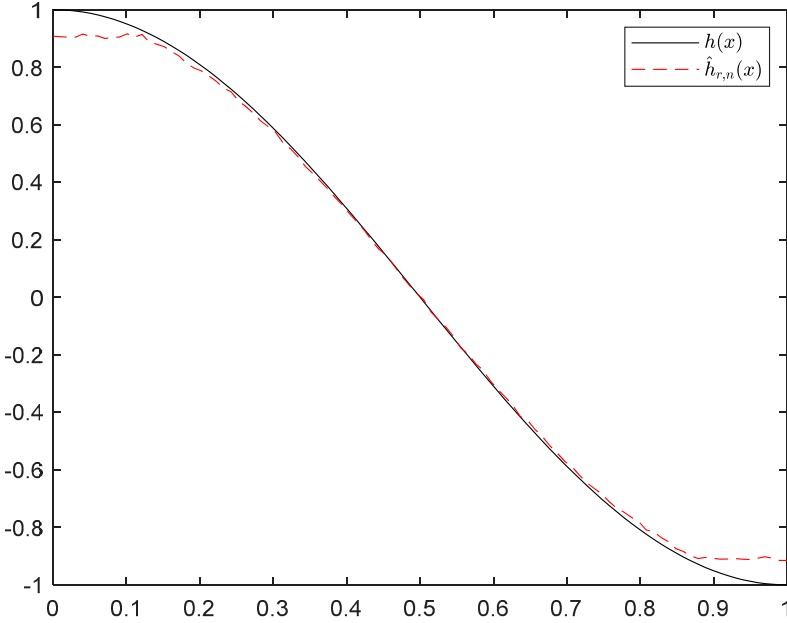

**Figure 18.** Curves of $h(x) = \cos(\pi x)$ and $\hat{h}_{r,n}(x)$ with $\beta = 3.5$ and $n = 1200$.

**Table 3.** The MSEs of $\hat{\beta}_{r,n}^{(LS)}$ with $\beta = 3.5$ and $h(x) = \cos(\pi x)$.

| $x$ | $n = 100$ | $n = 600$ | $n = 1200$ | $n = 1900$ | $n = 2700$ | $n = 3600$ |
|-----|-----------|-----------|------------|------------|------------|------------|
| 0.2 | 0.028735 | 0.0046935 | 0.002547 | 0.0017209 | 0.0010944 | 0.00085433 |
| 0.4 | 0.032703 | 0.0053621 | 0.0022595 | 0.0015934 | 0.001206 | 0.0008074 |
| 0.6 | 0.027853 | 0.0048229 | 0.0025756 | 0.0014096 | 0.0011848 | 0.00084765 |
| 0.8 | 0.03042 | 0.004848 | 0.002352 | 0.0014649 | 0.0011824 | 0.00084588 |

**Table 4.** The MSEs of $\hat{h}_{r,n}(x)$ with $\beta = 3.5$ and $h(x) = \cos(\pi x)$.

| $x$ | $n = 100$ | $n = 600$ | $n = 1200$ | $n = 1900$ | $n = 2700$ | $n = 3600$ |
|-----|-----------|-----------|------------|------------|------------|------------|
| 0.2 | 0.025536 | 0.0068733 | 0.0035259 | 0.0025282 | 0.0019082 | 0.0015027 |
| 0.4 | 0.029859 | 0.0061945 | 0.0032347 | 0.0022321 | 0.002002 | 0.0016778 |
| 0.6 | 0.02415 | 0.005475 | 0.0040859 | 0.0026873 | 0.0017436 | 0.0014589 |
| 0.8 | 0.027164 | 0.0055904 | 0.0036775 | 0.0024326 | 0.0017396 | 0.0014935 |

It can be seen from Figures 1–8 and Figures 10–17 that regardless of the values of $x$, $\hat{\beta}_{r,n}^{(LS)} - \beta$ and $\hat{h}_{r,n}(x) - h(x)$ fluctuate to zero and the ranges of $\hat{\beta}_{r,n}^{(LS)} - \beta$ and $\hat{h}_{r,n}(x) - h(x)$ decrease as $n$ increases. From Tables 1–4, one can see that regardless of the values of $x$, the MSEs decrease gradually as $n$ increases. Hence the estimators get closer and closer to their real values as $n$ increases. Figures 9 and 18 further show that the estimators of function $h(x)$ have good effects. The simulation results directly reflect our theoretical results.

## 7. Conclusions

In this paper, we mainly investigated the asymptotic properties of the estimators for the unknown parameter and non-parametric component in the heteroscedastic partially linear model (1). A lot of authors have derived the asymptotic properties of the estimators in partially linear models with independent random errors (see [4–6,8,33]). However, in many applications, the random errors are not independent. Here, we assumed that the random errors are $\rho^-$-mixing, which includes independent, NA, and $\rho^*$-mixing random variables as special cases. Under some suitable conditions, the strong consistency and $p$-th $(p > 0)$ mean consistency of the LS estimator and WLS estimator for the unknown parameter $\beta$ were investigated, and the strong consistency and $p$-th $(p > 0)$ mean consistency of the

estimators for the non-parametric component $h(\cdot)$ were also studied. The results obtained in this paper include the corresponding ones of independent random errors, NA random errors (see [16]), and $\rho^*$-mixing random errors as special cases. Furthermore, for the model (1), we carried out simulations to study the numerical performance of the asymptotic properties for the estimators of the unknown parameter and non-parametric component for the first time. $\rho^-$-mixing sequences are widely used dependent sequences. Therefore, investigating the limit properties of the estimators in regression models under $\rho^-$-mixing errors in future studies is an interesting subject.

**Author Contributions:** Methodology, Software, Writing—original draft, and Writing—review and editing, Y.Z.; Funding acquisition, Supervision, and Project administration, X.L.; Validation, Y.Z. and X.L. All authors have read and agreed to the published version of the manuscript.

**Funding:** This work was supported by the National Natural Science Foundation of China (61374183) and the Project of Guangxi Education Department (2017KY0720).

**Conflicts of Interest:** The authors declare no conflict of interest.

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
