# Peer review of "The Consistency of Estimators in a Heteroscedastic Partially Linear Model with ρ-Mixing Errors"

_symmetry, doi:10.3390/sym12071188_

Round 1

Reviewer 1 Report

Review of "Strong consistency and mean consistency of estimators in a heteroscedastic partially linear model with p-mixing errors"

This paper presents results on the strong consistency and the p-th mean consistency of least squares estimators and weighted least squares estimators for the heteroscedastic partially linear regression model for which the errors are rho^{-}-mixing random variables.

This is the first paper in the literature, which presents the strong consistency and the p-th mean consistency of least squares estimators and weighted least squares estimators for this model.

The paper is very-well written and clearly explains the contributions of the work and relates very-well the results to the body of literature. From Sections 2 to 5 the theoretical results are carefully demonstrated. In Section 6, a numerical simulation exercise supports the theoretical results of the paper.

Author Response

Responses to Reviewer 1 on symmetry-826098

Title: The consistency of estimators in a heteroscedastic partially linear model with rho^{-}-mixing errors

Comments: This paper presents results on the strong consistency and the p-th mean consistency of least squares estimators and weighted least squares estimators for the heteroscedastic partially linear regression model for which the errors are rho^{-}-mixing random variables.

This is the first paper in the literature, which presents the strong consistency and the p-th mean consistency of least squares estimators and weighted least squares estimators for this model.

The paper is very-well written and clearly explains the contributions of the work and relates very-well the results to the body of literature. From Sections 2 to 5 the theoretical results are carefully demonstrated. In Section 6, a numerical simulation exercise supports the theoretical results of the paper.

Responses:Thank you very much for your positive evaluations and approval for my manuscript.

Reviewer 2 Report

The title summarize the manuscript and the the abstract is well structured.

The paper is well organized, the methods are described in sufficient detail to understand the approach used and the results povided seem to bring new contributions for the literature on this topic, therefore I recommend its publication after the following minor typos:

Line 32: delete one of the  “the” ;

Line 103:  delete one of the  “some”;

In the references:

In [6] please italicize “ Stat. Papers “;

In [7] delete one of the 2018;

In [26] delete one of the 2015;

Author Response

Responses to Reviewer 2 on symmetry-826098

Title: The consistency of estimators in a heteroscedastic partially linear model with rho^{-}-mixing errors

Thank you very much for your insightful comments and suggestions. I have carefully revised the manuscript by incorporating all points raised by you and highlighted the changes in the revised manuscript. My point-by-point responds to your comments are as follows:

Comment 1: The title summarize the manuscript and the the abstract is well structured.

The paper is well organized, the methods are described in sufficient detail to understand the approach used and the results povided seem to bring new contributions for the literature on this topic, therefore I recommend its publication after the following minor typos:

Response 1: Thank you very much for your positive evaluations and approval for my manuscript.

Comment 2: Line 32: delete one of the “the” ;

Response 2: I have deleted the redundant “the” in line 32.

Comment 3: Line 103 (Line 82 in the revised manuscript): delete one of the “some”;

Response 3: I have deleted the redundant “some” in line 82.

In the references:

Comment 4: In [6] please italicize “ Stat. Papers “;

Response 4: I have italicized “ Stat. Papers” in [6].

Comment 5: In [7] delete one of the 2018;

Response 5: I have deleted the second 2018 in [7].

Comment 6: In [26] ([29] in the revised manuscript) delete one of the 2015.

Response 6: I have deleted the second 2015 in [29].

I greatly appreciate the Editor and Reviewer for your kind help and effort, and would hope that the corrections will meet with approval.

Reviewer 3 Report

See the attached document

Round 2

Reviewer 3 Report

Line 56: In the definition of \rho(T,U), the parentheses are still not in the right position; Should be E(XY)-E(X)E(Y) instead of E(XY)-(EX)(EY)

Line 77: Should be "The concept ... is .." instead of "The concept ... are .."

Line 92 & 93: The sentence in its current form form is not well formulated. The authors should change the sentence into:" The results obtained in the paper deal with independent errors as well dependent errors as special cases".

Line 138: C1-C4 were previously proposed by some other authors. For sake of clarity and completeness, this author should explain their meaning in the context of his own paper. Saying that they were previously used by some authors does not explain what they mean in the present context. Conditions imposed on models are generally explained in the context of the each paper, more so in statistics. 

Line 418: Figure 18. The author would be well served if a legend was added to plot. Moreover, in the caption, the author should change "full line" to continuous curve  and imaginary line to dotted curve.
